

# Genetic effects on life-history traits in the Glanville fritillary butterfly

Anne Duplouy[1,*], Swee C. Wong[1,*], Jukka Corander[2,3], Rainer Lehtonen[4] and Ilkka Hanski[5]

[1] Department of Biosciences, Metapopulation Research Centre, University of Helsinki, Helsinki, Finland
[2] Department of Mathematics and Statistics, University of Helsinki, Helsinki, Finland
[3] Department of Biostatistics, Institute of Basic Medical Sciences, Faculty of Medicine, University of Oslo, Oslo, Norway
[4] Institute of Biomedicine and Genome-Scale Biology Research Program, Biomedicum, Faculty of Medicine, University of Helsinki, Helsinki, Finland
[5] Department of Biosciences, University of Helsinki, Helsinki, Finland
* These authors contributed equally to this work.

Corresponding author
Anne Duplouy,
anne.duplouy@helsinki.fi

## ABSTRACT

**Background:** Adaptation to local habitat conditions may lead to the natural divergence of populations in life-history traits such as body size, time of reproduction, mate signaling or dispersal capacity. Given enough time and strong enough selection pressures, populations may experience local genetic differentiation. The genetic basis of many life-history traits, and their evolution according to different environmental conditions remain however poorly understood.

**Methods:** We conducted an association study on the Glanville fritillary butterfly, using material from five populations along a latitudinal gradient within the Baltic Sea region, which show different degrees of habitat fragmentation. We investigated variation in 10 principal components, cofounding in total 21 life-history traits, according to two environmental types, and 33 genetic SNP markers from 15 candidate genes.

**Results:** We found that nine SNPs from five genes showed strong trend for trait associations (*p*-values under 0.001 before correction). These associations, yet non-significant after multiple test corrections, with a total number of 1,086 tests, were consistent across the study populations. Additionally, these nine genes also showed an allele frequency difference between the populations from the northern fragmented versus the southern continuous landscape.

**Discussion:** Our study provides further support for previously described trait associations within the Glanville fritillary butterfly species across different spatial scales. Although our results alone are inconclusive, they are concordant with previous studies that identified these associations to be related to climatic changes or habitat fragmentation within the Åland population.

# INTRODUCTION

Human land use has led to natural habitat loss and fragmentation such that numerous species and populations are now at risk of extinction (*Hanski, 2005*;

*Hughes, Daily & Ehrlich, 1997*). The impacts of the remaining patch matrix structure or metapopulations, of the size and the number of remaining suitable habitat patches, of the level of connectivity or the isolation degree between the patches on the persistence and evolutionary dynamics of species and populations have been intensely investigated (*Hanski & Mononen, 2011*; *Hanski & Ovaskainen, 2000*). Life-history traits such as body size, reproductive output or dispersal capacity are known to often vary accordingly to the degree of fragmentation experienced by the different populations (*Hanski, 2011*; *Saastamoinen, 2007b*; *Wheat et al., 2011*). Although the type and the strength of natural selection in fragmented habitat might vary locally, given enough time and strong enough selection pressures, persisting populations might adapt to fragmentation and experience genetic differentiation. To date, the evolution of the genetic basis of many life-history traits according to habitat fragmentation remain however poorly understood. This also remains true in the case of the Glanville fritillary butterfly, *Melitaea cinxia* (L.), despite that the ecology of the species has been intensively investigated over the last 20 years in the Åland Islands, and the species is now considered as a model organism in the dynamics and evolution of natural (meta)population.

Allelic variation in metabolic genes is often associated with life-history traits in natural populations (*Marden, 2013*). A much-studied example is the gene *Phosphoglucose isomerase* (*Pgi*) in butterflies but also in other insects (*Dahlhoff & Rank, 2000*) and plants (*Mitchell-Olds & Pedersen, 1998*). In the Orange Sulfur butterfly (*Colias eurytheme*, Pieridae), *Pgi* allozyme alleles are associated with survival in the field (*Watt, 1977*) and reproductive success (*Watt, Carter & Blower, 1985*), while in the Glanville fritillary butterfly (*M. cinxia*, Nymphalidae), a SNP in *Pgi* is associated with flight metabolic rate (*Niitepõld, 2010*; *Niitepõld et al., 2009*) and dispersal rate in the field (*Hanski & Mononen, 2011*; *Niitepõld et al., 2009*), population growth rate (*Hanski & Saccheri, 2006*), female fecundity (*Saastamoinen, 2007b*), body temperature at flight (*Saastamoinen & Hanski, 2008*), lifespan (*Orsini et al., 2009*; *Saastamoinen, Ikonen & Hanski, 2009*) and larval development (*Saastamoinen et al., 2013*). Considering insect studies on other candidate genes, *Saastamoinen et al. (2013)* found genetic associations between the incidence of an extra larval instar and allelic variation in *Pgi*, *Serpin-1*, and *Vitellin-degrading protease precursor* genes. Importantly, many of these associations involve a significant interaction between genotype and ambient (*Niitepõld, 2010*) or acclimation temperature (*Wong et al., 2016*), consistent with the hypothesis of temperature-dependent enzyme kinetics.

Apart from association studies with candidate genes, genetic effects on life-history traits in the Glanville fritillary are demonstrated by significant heritabilities of many traits (*de Jong et al., 2014*; *Klemme & Hanski, 2009*; *Kvist et al., 2015*; *Mattila & Hanski, 2014*). For instance, recent studies have shown high heritabilities for both post-diapause larval development time ($h^2 = 0.37$) and lifetime offspring production ($h^2 = 0.62$) (*de Jong et al., 2014*; *Kvist et al., 2013*). These studies do not discriminate between single gene and multigene effects, though *de Jong et al. (2014)* found that genetic polymorphism in the cytochrome P450 gene *CYP337* accounted for 14% of the estimated heritability in egg production.

A generic problem with many candidate gene studies on wild populations is relatively small sample sizes and lack of replication, which cast doubt on the conclusions even in the case of significant results. Exceptions that demonstrate convincing single-locus associations include the study of the *Sdhd* gene polymorphism (*Sdhd* is a subunit of a *Succinate dehydrogenase*, SDH, gene), which is associated with a reduction in SDH enzyme activity in flight muscles and an increase of metabolic rate during flight in the Glanville fritillary (*Marden et al., 2013*; *Wheat et al., 2011*). Variation in the same locus is associated with longevity in hyperoxic conditions in *Drosophila* (*Walker et al., 2006*).

Here, we aimed to test whether habitat fragmentation might drive genetic differentiation between five natural populations of the Glanville fritillary butterfly, using numerous SNP marker-phenotypic trait associations. We show evidence of associations at various loci, through a candidate gene study involving a total of 49 single nucleotide polymorphisms (SNPs) from 26 candidate genes (metabolic genes). All SNPs were picked mostly based on previous similar, but more limited studies in terms of total number of traits, populations, or markers included. We investigated 21 different life-history traits, selected from traits previously recorded by a study by *Duplouy, Ikonen & Hanski (2013)* investigating phenotypic differentiation of butterfly populations under landscape fragmentation, and that we found to have high importance in three principal components (PCs) analyses. Importantly, the present study is replicated by including butterfly samples from five different regional populations in the Baltic area, one population in Finland, three in Sweden, and one in Estonia. Additionally, we also provide data from a pilot study, providing a temporal semi-replication to the main study (only on larval and pupal development traits). Several studies have shown that some differences in life-history traits (*Duplouy, Ikonen & Hanski, 2013*), allelic frequencies (*Fountain et al., 2016*) and gene expression profiles (*Somervuo et al., 2014*) occur between the populations inhabiting fragmented versus continuous landscapes. As both northern regional populations inhabit a highly fragmented landscape, including the well-studied metapopulation in the Åland Islands (*Hanski, 2011*), while the three remaining southern regional populations inhabit a more continuous habitat, we incorporated an environment factor confounding both the degree of habitat fragmentation and the latitude (or climatic region) of the populations, to our analyses to test if and how such factor may affect the Glanville fritillary butterfly at the genetic level. Although other major population characteristics, such as the different degree of relatedness between the populations, might play a role, our results present various potential genetic associations that despite being non-significant after accounting for all tests included in the study ($N = 1,086$) are consistent across the populations, and the two environments.

## MATERIALS AND METHODS

### Study material

We used the material reported in the phenotypic life-history study of *Duplouy, Ikonen & Hanski (2013)*. We included the results from a pilot study that took place in 2007, as they provide independent replicate to the findings from the main 2010 study. Sample sizes for each year in given in Table S1.

Pre-diapause larvae of the Glanville fritillary were sampled from five regional populations in the Baltic Sea region (Fig. 1) in September 2006 and September 2009. Both the Åland Islands (ÅL) metapopulation in Finland (60°07′N, 19°54′E) and the Swedish east-coast Uppland (UP, 59°90′N, 17°50′E) population occur in highly fragmented landscapes, where the meadow habitat occurs in small fragments (average area: 0.19 ha in ÅL, *Ojanen et al., 2013*), which cover only about 1% of the total landscape area (*Hanski, 2011*). In contrast, in Öland (ÖL, 56°39′N, 16°38′E) and Gotland (GO, 57°47′N, 18°49′E) in Sweden, and in Saaremaa (SA, 58°05′N, 22°15′E) in Estonia, there are extensive continuous calcareous meadows (alvars), often exceeding several hundred hectares in size. The larval host plants *Plantago lanceolata* and *Veronica spicata* are the same in all five populations (*Duplouy, Ikonen & Hanski, 2013*; *Nieminen, Siljander & Hanski, 2004*). Females lay large clutches of ca 150 eggs and the larvae live in groups of full sibs (*Nieminen, Siljander & Hanski, 2004*). Despite generally colder monthly temperature minima recorded across the years for both northern fragmented populations (Fig. S1), the butterfly phenologies are generally considered similar between the five populations of this study (*Duplouy, Ikonen & Hanski, 2013*). In any case, we confounded both the degree of fragmentation and the latitude of the population into one environment factor: northern fragmented environment versus southern continuous environment. For a more detailed description of the species' ecology and eco-evolutionary dynamics see *Nieminen, Siljander & Hanski (2004)* and *Hanski (2011)*, respectively.

For the pilot study ran in the summer 2007, our samples originated from different larval family groups in each population ($N_{\text{ÅL}} = 9$, $N_{\text{UP}} = 13$, $N_{\text{GO}} = 2$, and $N_{\text{SA}} = 12$), with several individuals from the same family included. In contrast, for the experiment ran in the summer 2010, our samples originated from 50 larval family groups from each population, widely distributed across the respective study areas, and for which the degree of relatedness was previously estimated to be low (estimate of genetic similarity by *Duplouy, Ikonen & Hanski (2013)*). Most butterflies belong to the eastern mitochondrial clade, originating from the Asian glacial refuge area, with the minor exception of a few butterflies from ÖL belonging to the southeastern clade (*Duplouy, Ikonen & Hanski, 2013*; *Wahlberg & Saccheri, 2007*). According to the IUCN Red List, the butterfly is not classified as a threatened species in the five study regions, hence no permits were required for sampling.

For a detailed description of the rearing conditions and measuring of the phenotypic traits, see *Duplouy, Ikonen & Hanski (2013)*. In brief, the caterpillars diapaused in the constant temperature of 3 °C in incubators, the post-diapause larvae were reared indoors under 12:12 L/D light and 25/15 °C day/night temperature and were fed with the leaves of *P. lanceolata*, cultivated in the laboratory. Adult butterflies were individually marked and released into an outdoor population cage to collect data on behavioral and reproductive traits as described below.

## Phenotypic traits
### Larval and pupal development
For each individual in the 2007 pilot study, we only recorded the sex, the pupal weight (mg) and the post-diapause larval period (from fifth instar molt to pupation, in days).

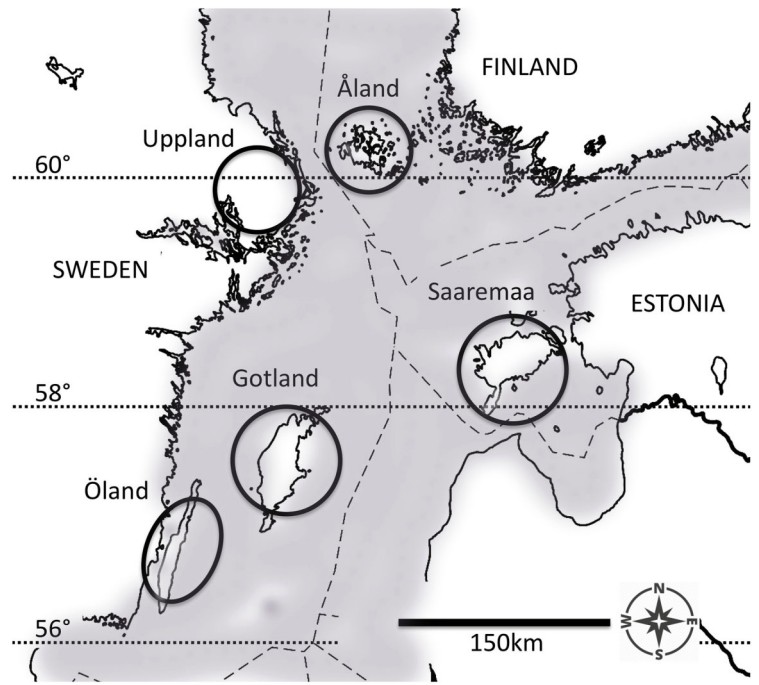

**Figure 1 The five populations of the Glanville fritillary butterfly in the Baltic Sea region.** The Finnish population on the Åland Islands, and the Swedish population on the coastal region of Uppland inhabit northern fragmented environments. The Swedish populations on the Öland and Gotland Islands, and the Estonian population on the Saaremaa Island evolve in southern continuous environments. The map was created by A. Duplouy using a modified version of the image "Location map of the Baltic Sea" (https://commons.wikimedia.org/wiki/File:Baltic_Sea_location_map.svg, under the license CC BY-SA 3.0, https://creativecommons.org/licenses/by-sa/3.0/) by NordNordWest/Wikipedia. The original image was cropped using Adobe Photoshop CS6 (Version: 13.06 × 64, http://www.adobe.com/products/photoshop.html).

For each individual collected in the fall 2009, we recorded the sex and the weights of the fifth, sixth, and seventh instar larvae and pupae, the time period between successive molts, and the pupal period.

### Outdoor cage experiment

We did not collect any adult traits from the 2007 pilot study. Twenty-four hours after eclosion, butterflies emerging from the larvae collected in the fall 2009, were marked with a unique number written on the hind wing and released into a large outdoor population cage (32 × 26 × 3 m) at the Lammi biological station in southern Finland (*Duplouy, Ikonen & Hanski, 2013*; *Hanski, Saastamoinen & Ovaskainen, 2006*; *Saastamoinen, 2008*). Data on each mating and oviposition were collected by monitoring the butterflies continuously between 8 am and 6 pm for 10 sunny days ranging between the 4th and the 20th of June 2010. A sunny day is defined as a day with no or little overcast, thus allowing the warming-up of the butterflies' body temperature and the initiation of activity in the cage (flight and oviposition are initiated at ambient temperature $T > 13\,°C$, *Saastamoinen, 2007a*, *2007b*; *Saastamoinen & Hanski, 2008*). The temperatures experienced in June 2010 ranged from 11 °C to 23.5 °C (monthly average = 16.6 °C),

which remains well within the range of the average temperatures recorded in the five populations (Fig. S1). At 9, 11, 13, 15, and 17 h, we recorded the position in the cage (for a map of the cage see: *Duplouy, Ikonen & Hanski, 2013*), and whether the encountered butterfly was flying or not (probability of the butterfly flying in the cage). The average distance between two observations of the same individual was calculated as the cumulative distance between all consecutive observations divided by the total number of observations for the butterfly. For both males and females, we calculated the age at first mating and mating success (total numbers of observed copulations). The survival of each individual was defined as the time between eclosion and its last observation, dead or alive.

The cage had 200 potted host plants, 100 plants of *P. lanceola* and *V. spicata* each. Each egg clutch laid by a female was placed individually on a petri dish labelled with the female identity, the host plant species, and the date of oviposition, and incubated in the laboratory. The host plants were monitored for egg-laying females intensively, to record each egg clutch (*Duplouy, Ikonen & Hanski, 2013*). Eggs were counted three days after oviposition and the caterpillars were counted three days after eclosion. For each female, we calculated the age at first oviposition, total number of clutches, eggs, and caterpillars, the host plant preference (the proportion of clutches laid on *P. lanceolate*), and the egg hatch rate of each clutch (percentage of eggs that hatched).

As many of the traits recorded were highly intercorrelated (Table S2, Pearson's correlation), and to avoid the use of a large number of tests and the subsequent increased risk of type I error, we reduced our analyses to three independent principal component analyses (PCAs), and 10 PCs. Chosen PCs all had eigen-values over 1, and proportion of variance values over 0.1. The first PCA ($PCA_1$) included all the larval and pupal traits from both sexes using only those individuals for which information was available for all developmental traits (as described in *Duplouy, Ikonen & Hanski, 2013*). $PCA_1$ provided four PCs ($PC_{1-1}$ to $PC_{1-4}$). The second PCA ($PCA_M$) included only male adult traits and produced three PCs ($PC_{M1}$ to $PC_{M3}$), while the third PCA ($PCA_F$) included only female adult traits, and produced three PCs ($PC_{F1}$ to $PC_{F3}$) (see Tables S3–S5 for details on each PCAs, and description of the phenotypic traits included in each PCA). The probability of a butterfly being observed flying, the hatch rate of all clutches and of the first clutch only, and the host plant preference were treated as proportions, and were ArcSin-transformed prior being integrated into the PCAs. The 10 analyzed PCs accounted for the majority of the total variation in the traits included in the respective PCAs (74% in developmental-traits, and over 58% in male and female adult-traits). Normality of the ten PCs was assessed using Shapiro–Wilk tests ($\alpha = 0.01$).

## Genotyping

A small fragment of the hind wing (3.14 mm$^2$) was sampled from each butterfly using sterile biopsy punch (Tamro, Vantaa, Finland). The biopsy does not impair the flight capacity of the butterfly (*Hanski, Saastamoinen & Ovaskainen, 2006*). Wing tissues, plus all body tissues from individuals recovered at the end of the experiment, were stored individually in 99.6% ethanol. The DNA was extracted at the Institute of Biotechnology, University of Helsinki, using Nucleo Spin® 96 Tissue kit (Macherey-Nagel GmbH & Co.

KG, Düren, Germany) following manufacturer's protocol. DNA was amplified using the Illustra GenomiPhi V2 DNA Amplification Kit according to manufacturer's protocol (GE Healthcare, Piscataway, NJ, USA). All DNA extracts were genotyped for two SNPs from the *Pgi* gene, or all 49 SNPs from 26 candidate genes, for the pilot and main experiment samples, respectively (Table S6).

The candidate genes were selected based on a study by *Ahola et al. (2015)*, which compared transcriptomic variation between individuals from Åland Islands and China, and revealed genes with significantly reduced variation in the Åland Islands, possibly due to a selective sweep or local adaptation. The resulting gene set was tested by gene set enrichment analysis. Among the overrepresented GO groups, genes belonging to serine proteases, to serine protease inhibitors and to cuticular proteins GO groups were chosen for genotyping (*Ahola et al., 2015*). Following selection, 31 SNP markers from nine genes fulfilled the quality criteria and were retained for further analysis. To this list, we added the following markers: (i) three SNPs from the *Phosphoglucose isomerase* (*Pgi*) gene (*Orsini et al., 2009*), (ii) eight SNPs with minor allele frequency (MAF) >0.2 in an annotated EST library of *Pgi* pathway associated genes (upstream and downstream) (*Ahola et al., 2015*), and (iii) 10 SNPs with interesting expression patterns in a chip expression dataset (*Kvist et al., 2013*). Table S6 provides the list of the all selected markers. All SNPs included in this study were bi-allelic SNPs. SNP genotyping was performed at the Institute for Molecular Medicine Finland (FIMM, Helsinki, Finland) using Sequenom iPLEX Gold chemistry. The results were validated for seven independent samples by direct genomic sequencing with AB 3730 (Applied Biosystems Europe, Bleiswijk, The Netherlands). We used MassARRAY Assay Design 3.1 to design the primers for Sequenom (San Diego, CA, USA), while the validation primers and extension probes were designed using Primer3 (*Rozen & Skaletsky, 2000*).

The quality of SNPs, including deviation from the Hardy–Weinberg equilibrium (HWE), was assessed using an in-house quality control pipeline (*Wong, 2011*). False discovery rate calculation was used to correct for multiple testing ($\alpha = 0.05$) (*Hochberg & Benjamini, 1990*). For the pilot study, one of the two *Pgi* SNPs genotyped did not reach our quality criteria, while for the samples collected in 2009, out of the 49 SNPs genotyped, 16 did not reach the quality criteria; they were hence excluded from subsequent analyses. The criteria included unclear heterozygote/homozygote cluster patterns and no heterozygote cluster. We also remove SNPs with significant deviations from the HWE or a MAF <0.05, or both in two or more of the populations. Similarly, if only one of the four populations did not comply with the HWE or the MAF at a certain SNP, we removed that population from the analyses (Table S6).

## Data analyses

### Phenotype–genotype associations

Subsequent analyses were carried out using R 3.1.3 software (*R Development Core Team, 2013*). For the 2007 pilot study, we used linear mixed models to analyze separately the effect of the *Pgi* SNP (*Pgi*:c.331A>C) on the pupal weight and post-diapause larval period. Sex, environment type, and genotype were used as explanatory variables.

Population and family were added as a random factors nested within environment in the models.

For the samples collected in 2009, we used linear mixed models to separately test the effect of 33 SNPs on the four PCs ($PC_{1-1}$ to $PC_{1-4}$) describing post-diapause larval and pupal development. Sex, environment type, genotype, and environment by genotype interaction were used as explanatory variables to those models, while population was included as a random factor nested within the environment type. Similar models, excluding the sex factor were used to test genetic associations to the three PCs from both $PCA_F$ and $PCA_M$. Initial models with interaction term were subsequently simplified by the removal of non-significant terms to give a final minimal adequate model.

We accounted for additive, dominant, recessive, and over-recessive/dominant effects of the alleles, and conducted false discovery rate calculation (*Hochberg & Benjamini, 1990*) to correct for multiple testing for all tests. This means that each raw *p*-value was corrected for a total of 1,086 tests conducted (including four inheritance models × 10 PCs × 33 SNPs that reached our quality criteria, Table S6). A FDR adjusted *p*-value of 0.05 means 5% of the uncorrected significant discovery will result in false positives.

In the Glanville fritillary, the linkage disequilibrium (LD) between genes ($r^2$) is currently estimated to reach a level of 0.45 at about 300 bp distance (*Ahola et al., 2014*; *Rastas et al., 2013*), but may extend over thousands of base pairs across the genome of the butterfly ($r^2 = 0.3$), based on data from the Åland Island only). For simplicity of the analysis, we decided to investigate a 300 bp window from both ends of each gene with significant association with phenotypic trait(s), thus to identify neighboring genes and functional variants (*Ahola et al., 2014*) potentially linked to the marker SNPs. Correlation between SNPs (LD) was not taken into account in the genetic association analyses. We also performed power analyses using genetic power calculations (*Purcell, Cherny & Sham, 2003*), including the allele frequency of the tested SNP (0.3), the allele frequency of the causal SNP (0.3), the LD between target and causal SNP (0.1–0.5), our sample size ($N = 171$) and the estimated effect size for either small or large effect (total QTL = 0.1 or 0.5, respectively), and for both additive and dominant models.

### Allele frequency

A directed permutation test was used to assess the strength of evidence for environment type having influenced the allele frequencies at a particular SNP. To account for multiple testing, we applied a conservative Bonferroni correction at 1% significance level on the nominal *p*s derived using 1,000,000 random permutations of the sample labels. Given the data at a locus, let $p_{11}$, $p_{12}$ be the maximum likelihood estimates of the MAF in the northern fragmented populations (ÅL, UP). Let $p_{21}$, $p_{22}$ be the corresponding estimates in the southern continuous populations (ÖL/GO, SA). For each locus, the test statistic was calculated as the vector of MAF differences: $p_{11} - p_{21}$, $p_{11} - p_{22}$, $p_{12} - p_{21}$, and $p_{12} - p_{22}$. Thus, signs and absolute values of the elements of the test statistic reflect collectively whether the MAFs are consistently smaller or greater in the northern fragmented environment, and how large the differences are. For every locus in which the differences were all consistently positive or negative, the random permutations were used

to calculate the probability that at least as large a difference in the same direction was observed by chance, which resulted in the nominal empirical *p*-values for the locus. As the allele frequency differences between environments may be confounded by the distance separating the populations (isolation by distance), we measured pairwise genetic distances between populations (FSTs, $\alpha = 0.05$) using the allelic profiles of the 33 SNPs that fulfilled our quality criteria, and the software GenoDive 2.0b27 (*Meirmans & Van Tienderen, 2004*).

Additionally, the genotypes of 36 SNPs (33 SNPs that passed our quality criteria and three extra SNPs) were encoded into numerical form. We then implemented the "heatmap.2" function from the gplots R package (*Warnes et al., 2009*) with the numerical data from either all SNPS (Fig. S2A), or only from the 15 SNPs identified as showing significant allele frequency differences between environment types using direct permutation tests (Fig. S2B). In the heatmap.2, the relative count of alleles for each SNP and each sample is calculated to allow the hierarchical clustering of the samples based on their allele count. Samples similarity is represented as a dendogram on the left side of each heatmap.

### Data accessibility

All data are available in the public archive *Dryad* under the provisional data identifier: doi 10.5061/dryad.81r20. Temperature data for the 1992–2001 period from Uppland coastal region, Öland and Gotland were extracted from the Metadata file ISO 19115:2003 ECDS (file identifier: 7085c74e-58a4-49f0-b8c0-b5a0eb3c74f9) publicly available from the Swedish Meteorological and Hydrological Institute (SMHI) website (http://www.smhi.se/en). The publicly available temperature data from Åland and Saaremaa were provided by the Finnish Meteorological Institute (http://en.ilmatieteenlaitos.fi/), and extracted from the RIMFROST database (http://www.rimfrost.no), respectively.

## RESULTS

After correction, no significant results were conserved ($\alpha = 0.1$, all $p > 0.12$). Due to the large number of tests conducted and corrected for in this study, and to the relatively small sample size, our analyses might however have generated false negative results. Thus, we present below the results from the most significant associations before correction (Table 1; Table S7). In order to achieve 80% power, our sample size for testing a large effect SNP should be of 84 or 59, for dominant or additive model, respectively, when LD is of 0.5. However, as our SNPs are more likely to be small effect SNPs, to achieve 80% power, our sample size should be of 439 and 310 for dominant and additive models, respectively, with LD of 0.5. In all case, the sample size would increase to $N = 11,670$ for dominance model, and $N = 7,845$ for additive model, when the LD is small (LD = 0.1).

### Associations involving larval and pupal traits

The traits of larval and pupal development were summarized by the first four PCs of $PCA_1$ (Table S3) (*Duplouy, Ikonen & Hanski, 2013*). Of the four PCs, $PC_{1-1}$ is positively

**Table 1 Description of the SNP-phenotypic trait associations with lowest $p$-values before correction for multiple testing, with model of inheritance.** Both $p$-values from before and after FDR are given.

| Gene | SNP | Trait | Figure | Inheritance | $p$-value | FDR adjusted $p$-value |
|---|---|---|---|---|---|---|
| Pgi | (Pgi):c.331A>C | PC$_{1-3}$ | Fig. 2 | Dominant | 3.84e$^{-4}$ | 0.12 |
| Pgi | (Pgi):c.331A>C* | Pupal weight | Fig. S3B | Dominant | 0.0032 | 0.12 |
| SgAbd-8 | c480_est:926G>A | PC$_{1-2}$ | Fig. 3 | Recessive | 2.27e$^{-3}$ | 0.12 |
| SgAbd-8 | c480_est:1003G>C | PC$_{M2}$ | Fig. 4A | Dominant | 2.15e$^{-3}$ | 0.12 |
| SgAbd-8 | c480_est:926G>A | PC$_{M3}$ | Fig. 4B | Dominant | 8.56e$^{-4}$ | 0.12 |
| SgAbd-8 | c480_est:1051G>A | PC$_{M3}$ | Fig. 4B | Dominant | 8.56e$^{-4}$ | 0.12 |
| Hemolymph proteinase-5 | c50_est:735A>G | PC$_{1-2}$ | Fig. S4 | Over-dominant | 1.15e$^{-3}$ | 0.12 |
| Hemolymph proteinase-5 | c50_est:824A>G | PC$_{1-2}$ | Fig. S4 | Over-dominant | 1.21e$^{-3}$ | 0.12 |
| Hemolymph proteinase-5 | c50_est:824A>G | PC$_{M3}$ | Fig. S7B | Additive | 2.35e$^{-3}$ | 0.12 |
| Heat shock protein | hsp_1:106G>A | PC$_{1-1}$ | Fig. S5 | Over-dominant | 2.18e$^{-3}$ | 0.12 |
| Heat shock protein | hsp_1:206T>G | PC$_{M3}$ | Fig. S7A | Recessive | 1.33e$^{-3}$ | 0.12 |
| Serine proteinase-like protein | c3917_est:386A>C | PC$_{M1}$ | Fig. S6 | Recessive | 1.64e$^{-3}$ | 0.12 |

**Note:**
\* Identifies the significant result from the 2007 pilot study, while all other results are from the 2010 experiment.

correlated with pupal weight, but negatively correlated with pupal period. The other PC's show strongest correlations with the weights and durations of specific larval instars. The sex of the larvae had a strong significant effect on PC$_{1-1}$ and PC$_{1-3}$ ($p < 5.13e^{-6}$), with females showing lowest PC$_{1-1}$ and PC$_{1-3}$ values. Sex had no effect on neither PC$_{1-2}$ nor PC$_{1-4}$ (Table S7, *Duplouy, Ikonen & Hanski, 2013*).

One association was found between *Pgi*: c.331A > C and PC$_{1-3}$ ($p = 3.84e^{-4}$, dominant effect). Individuals with one or two copies of the C allele had higher values of PC$_{1-3}$ than the AA homozygotes (Fig. 2). PC$_{1-3}$ is primarily negatively correlated with the weights of the fifth instar post-diapause larvae (Table S3), thus suggesting that AA homozygotes tend to be heavier after diapause than the other genotypes (CC 4.24 mg, CA 4.04 mg, and AA 5.27 mg, Fig. S3A). This is true for all four populations, even if the frequency of the C allele is significantly higher in the southern continuous than in northern fragmented environments ($p < 10^{-6}$, Table S8). Similarly, the pupae with one or two copies of the C allele were heavier than the AA homozygotes in the 2007 pilot material (CC 192 mg, CA 193 mg, and AA 176 mg, $p = 0.0032$, dominant effect, Fig. S3B).

Both SNPs c480_est:926G>A and c480_est:1051G>A (125 bp's apart) in the *Endocuticle structural glycoprotein* gene (*SgAbd-8*) showed significant interactions with environment type in affecting PC$_{1-2}$ ($p = 0.0023$ and $0.002$, respectively, dominant effects, Fig. 3). There were no significant differences between the genotypes in the southern continuous environments ($p = 0.28$), but the individuals from the northern fragmented environments carrying at least one G allele had lower values of PC$_{1-2}$ than the AA homozygotes ($p = 0.019$, Fig. 3). PC$_{1-2}$ is mostly negatively correlated with the weight of the seventh larval instar (Table S3).

Finally, there was a significant effect of two SNPs in the *Hemolymph proteinase-5* gene (89 bp's apart) on PC$_{1-2}$ ($p = 1.15e^{-3}$, and $p = 1.21e^{-3}$, over-dominant effects, Fig. S4)
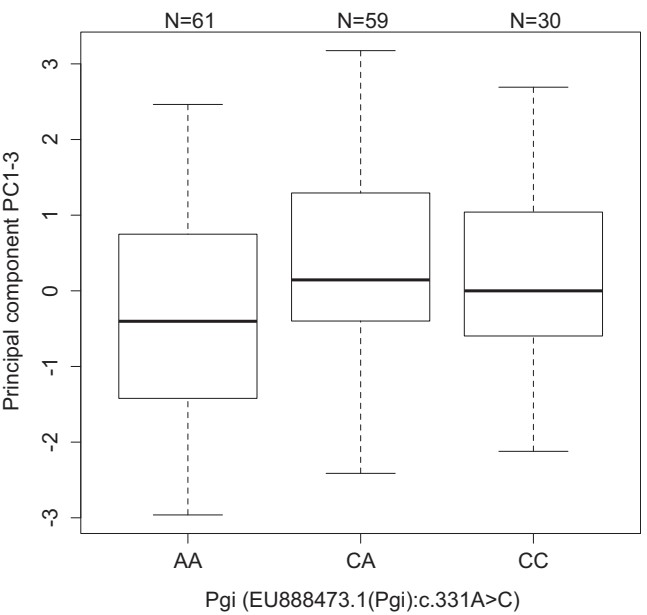

**Figure 2 Values of development PC1-3 for the different genotypes of the SNP *Pgi*:c.331A>C in the gene *Pgi*.** Sample size is given by the number above the bar. Heavy horizontal lines represent median values, boxes give interquartile ranges, whiskers and the dot give the minimum and maximum, and outlier values, respectively.

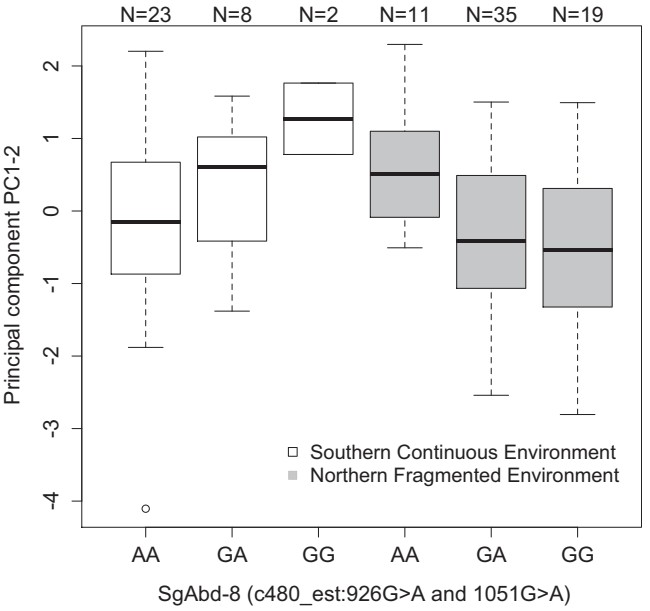

**Figure 3 Differences in the development PC$_{1-2}$ values for different butterfly genotypes in two SNPs in the *SgAbd-8* gene, evolving in southern continuous (white bars) or northern fragmented environment (gray).** Sample size is given by the number above the bar. Heavy horizontal lines represent median values, boxes give interquartile ranges, whiskers and dots give the minimum and maximum, and outlier values, respectively.

and a SNP in the *Heat shock protein* gene on $PC_{1-1}$ ($p = 2.19e^{-4}$, over-dominant effect, Fig. S5). In each case, the homozygotes show higher PCs values than heterozygote individuals.

## Associations involving adult traits

The male adult-traits were summarized by the first three PCs of $PCA_M$ (Table S4) to reflect on male quality. $PC_{M1}$ is negatively correlated with both survival and average distance flown per day, indicating that males of short survival do fly less. $PC_{M2}$ is negatively correlated with pupal weight but positively correlated with age at first mating, indicating that smaller males mate at older age. Finally, $PC_{M3}$ positively correlated flight activity (probability to fly) with mating success (total number of copulations and age at first mating), thus male flying more mate more, despite starting to mate at older age. There were no direct effects of environment on any of the PCs.

We found an association between $PC_{M1}$ and the SNP c3917_est:386A>C in the *Serine proteinase-like protein* gene ($p = 1.64e^{-4}$, recessive effect, Fig. S6). Butterflies with at least one copy of the A allele showed lower $PC_{M1}$ values (AA = −3.07, AC = 0.13, and CC = 15.09).

We found an association between SNPs in the *SgAbd-8* gene and $PC_{M2}$ (SNP c480_est:1003G>C, $p = 2.15e^{-4}$, dominant effect of G allele, Fig. 4A), and $PC_{M3}$ (SNP *c480_est:926G>A* and SNP *c480_est:1051G>A*, 125 bp's apart, $p = 8.56e^{-4}$, dominant effects of G allele). Additionally, the direct effects of the two SNPs in the *SgAbd-8* gene were coupled with the effect of environment by genotype interactions ($p = 1.87e^{-4}$, over-dominant effects, Fig. 4B). There were no significant differences between the genotypes in the northern fragmented environments, but the heterozygote individuals from the southern continuous environments had higher $PC_{M3}$ values than the homozygote individuals (Fig. 4B).

There was also a significant association between $PC_{M3}$ and the SNP hsp_1:206T>G in the *Heat shock 70 kDa protein* coding gene ($p = 1.33e^{-4}$, recessive effect, Fig. S7A). In general, individuals that carry at least one T allele show lower $PC_{M3}$ values (GG = 0.45, GT = −0.14, and TT = −0.07). Finally, there were significant associations between $PC_{M3}$ and the SNP c50_est:824A>G in the *Hemolymph proteinase-5* gene. Homozygote CC males show highest $PC_{M3}$ values (CC = 0.18, AC = −0.02, and AA = −0.16, $p = 2.35e^{-4}$, additive effect, Fig. S7B).

The female adult and oviposition related-traits were summarized by the first three PCs ($PC_{F1}$ to $PC_{F3}$) of a female $PCA_F$ (Table S5). $PC_{F1}$ is negatively correlated with female offspring production (both total eggs and total caterpillars production), suggesting that females laying small clutches produce less caterpillars. $PC_{F2}$ shows positive correlation with first clutch size and hatch rate (ArcSin corrected), suggesting that larger clutches show higher hatch rate success. Finally, $PC_{F3}$ is strongly positively correlated to female age at first oviposition, and to a lower degree to the size of the first clutch laid, suggesting that the first clutch of older females are larger. There were no effects of the environment nor the genotype on neither of the three PCs from $PCA_F$.

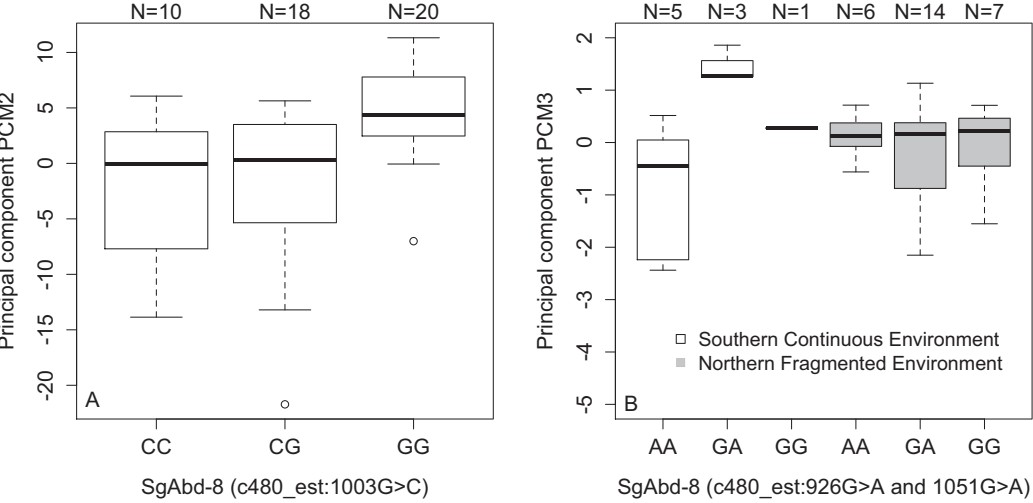

**Figure 4 Variations in PCs-values of males of the different genotypes in three SNPs in the *SgAbd-8* gene.** (A) PCM2 values per genotypes in the SNP c480_est:1003G>C. (B) Effect of environment on PCM3 values per genotypes in the SNPs c480_est:926G>A and c480_est:1051G>A. Individuals from southern continuous environment are shown in white, while individuals from northern continuous environment are shown in gray. Sample size is given by the number above the bar. Heavy horizontal lines represent median values, boxes give interquartile ranges, whiskers and dots give the minimum and maximum, and outlier values, respectively.

## Allele frequencies in northern fragmented versus southern continuous environments

There were 15 SNPs in eight genes with significant allele frequency differences between the northern fragmented and southern continuous environments, using directed permutation calculations. The eight genes are *Phosphoglucose isomerase* (*Pgi,* for both years), *Serine proteinase-like protein*-1, *Hemolymph proteinase-5, Vitellin-degrading protease precursor, Endocuticle structural glygoprotein* (*SgAbd-8*), *Cytochrome P450* (*CYP337*), *Heat shock 70 kDa protein* (*Hsp70*), and *Glucose-6-phosphate-1-dehydrogenase* (*G6PD*). Five of these genes also showed a significant association with larval development traits and/or male adult traits (as further described above), namely *Pgi, Hemolymph proteinase-5, Vitellin-degrading protease precursor, HSP70*, and *SgAbd-8* (Table S8). Additionally, pairwise FSTs (Table 2) revealed that the four populations used in the main experiment were not significantly different or similar to each other. We also did not identify any other genes or apparent functional variants within a 300 bp LD ($r^2 = 0.45$) window from spanning any of these eight genes, or any of the other genotyped SNPs. As there is no data available on the occurrence of epistasis in the Glanville fritillary butterfly, we also could not rule out the effect of hiding or masking gene(s) on the different associations tested in this study.

The hierarchical clustering of samples used to generate the heatmaps supports the evidence of environment-specific variation in allele count for several SNPs with significant allele frequency differences between the northern fragmented and southern continuous environments, when using the directed permutation calculations described above. The environment-specific differentiations illustrated in the heatmaps (Fig. S2) are especially

**Table 2 Pairwise FSTs (in bold) and pairwise standardized FSTs between the four populations collected in 2009, as calculated using GENODIVE (*Meirmans & Van Tienderen, 2004*).**

| Populations | ÅL | ÖL | SA | UP |
|---|---|---|---|---|
| Sample size (*N*=) | 35 | 39 | 58 | 44 |
| ÅL | – | **0.084** | **0.269** | **0.072** |
| ÖL | 0.130 | – | **0.170** | **0.164** |
| SA | 0.383 | 0.223 | – | **0.328** |
| UP | 0.115 | 0.251 | 0.459 | – |

Note:
ÅL, stands for the Finnish population in the Åland Islands; ÖL and UP, for the Swedish populations in the Öland Island and the Uppland coastal region, respectively; SA, for the Estonian population in the island of Saaremaa.

obvious for the SNPs in the *Serine proteinase-like protein*-1 gene, the *Hemolymph proteinase-5* gene, the *CYP377* gene, *the SgAbd-8* gene, the *Pgi* gene, and the *Glucose-6-phosphate-1-dehydrogenase* (*G6PD*) gene.

## DISCUSSION

We highlight several associations between SNPs in candidate genes and some of the PCs from PCAs on various life-history traits in the Glanville fritillary butterfly (Table 1), however these associations did not hold up following correction for multiple testing. One reason for the lack of significant results may remain the small sample size for each population, and the large amount of tests done, leading to false negative results. We therefore believe many of the positive results before correction, are robust and most likely driven by the fact that the selection of candidate genes was largely based on previous studies (discussed below), in which these genes had already shown significant or nearly significant effects. Furthermore, we used phenotypic traits, some of which have been highly heritable in previous studies (*de Jong et al., 2014*; *Kvist et al., 2013*) or have shown phenotypic variation among environments suggestive of local adaption (*Duplouy, Ikonen & Hanski, 2013*). Although functional validations are needed to confirm our findings, the present results provide a starting point for genome-wide association studies on the Glanville fritillary. Such studies have remained impractical so far, but they are becoming feasible in the future, with the help of the now available genome (*Ahola et al., 2014*), linkage map (*Rastas et al., 2013*) and other tools for the Glanville fritillary, as well as the decreasing cost of whole-genome sequencing.

The material for the present study originates from five distinct populations, over two different collection years. Consistent results across this material can be considered to be more robust than would be comparable results for a single population in the Glanville fritillary butterfly. However, because the phylogenetic relatedness varies between the five populations (see *Somervuo et al., 2014*, and Table 2 for estimates of population divergence; and *Wahlberg & Saccheri, 2007*, for description of the different mitochondrial clades), it is not possible to conclude with certitude that the genes with known or presumed function in life-history ecology show significant differences in allele frequencies due to variable natural selection in the two environments (northern fragmented versus southern continuous). Other non-investigated population characteristics, such as the

historical relatedness or geographical closeness of the populations from similar environments, might be of similar or higher importance in this context. Although the pairwise genetic distances between populations, calculated based on the genotypes at 33 SNPs of 176 butterflies are not significantly grouping the populations in the two environments, future association studies in the Glanville fritillary butterfly should further address this issue by including additional neutral markers. In any case, the fact that out of the 49 SNPs from 26 genes investigated, 15 SNPs from eight genes seem to show differences in allele frequencies between the two environmental extremes suggests their potential role in local habitat adaptation.

In each case, there is support from previous studies on the Glanville fritillary, or from other insect studies, and to some extend from the present study, that these genes may influence life-history traits or other phenotypic traits. For example, while working on four of the five populations included in the present study (Åland, Uppland, Öland, and Saaremaa), *Somervuo et al. (2014)* identified almost 2,000 genes with dissimilar expression levels between the northern fragmented and southern continuous environments. This list of genes in *Somervuo et al. (2014)* includes the eight genes for which we found diverging allele frequencies (at one or several SNPs) between environmental types. The key published results supporting our findings regarding the association of these eight genes to variation in life-history traits or other important phenotypic traits include the following:

1. The glycolytic *Phosphoglucose isomerase* (*Pgi*) gene, for which genetic polymorphism has previously been linked to variation in a large number of traits (*Dahlhoff & Rank, 2000*; *Hanski & Saccheri, 2006*; *Niitepõld et al., 2009*; *Orsini et al., 2009*; *Saastamoinen, Ikonen & Hanski, 2009*) as further discussed below, and for which our results support for associations with larval development.

2. The *Cytochrom p450* (*CYP337*) gene, which has been previously associated with life-time egg and larval production, and host plant preference in the Glanville fritillary (*de Jong et al., 2014*).

3. The *Vitellin-degrading protease precursor* gene, for which polymorphism has been associated with variation in larval development (*Ahola et al., 2015*, this study) and the incidence of an extra larval instar in the Glanville fritillary (*Saastamoinen et al., 2013*).

4. The *Heat shock 70 kDa protein* (*Hsp70*) gene, for which a previous study shown that gene expression is up-regulated during thermal stress (*Luo et al., 2014*; *Mattila, 2015*), as HSPs may provide cells with protection against damage from extreme temperature conditions (*Sørensen, Kristensen & Loeschcke, 2003*). Furthermore, in the Glanville fritillary, polymorphism in HSP genes have been previously associated with variation in fecundity traits, including egg hatch rate, age at first oviposition and first clutch size and in mobility (*de Jong et al., 2014*) and in male flight metabolic rate (*Mattila, 2015*). In the present study, polymorphism in the SNPs $hsp\_1{:}106G{>}A$ and $hsp\_1{:}206T{>}G$ are associated to variation in $PC_{1-1}$ and $PC_{M3}$ in males, respectively, thus suggesting association with larval development, or male flight activity and mating success.

5. The *Hemolymph proteinase-5* and the *Serine proteinase-like* genes are two genes linked to immune and stress responses in Lepidoptera (*An et al., 2009*; *Han et al., 2013*) and other insects (*Ashida & Brey, 1995*), and to larval development traits in the Glanville fritillary (*Ahola et al., 2015*, this study). Again, the correlations we observe in larvae and in males between $PC_{1-2}$ (larval development) or $PC_{M3}$ (male mating success and flight activity) and polymorphism in the *Hemolymph protease-5* gene might suggest variation in the quality, through for example variation in immune functions, between individuals in the Glanville fritillary.

6. The *Endocuticle structural glycoprotein* (*SgAbd-8*) gene, described as a structural gene associated with larval development in insects (e.g., in bumblebees, *Colgan et al., 2011*), showed genetic associations with male PCs, as well as a gene by environment type interaction on larval development in the present study.

7. The *glucose-6-phosphatase dehydrogenase* (*G6PD*) gene, in which polymorphism has been associated with mating success in *Colias* butterflies (*Carter, 1988*).

The results of the genetic association for the SNP (*Pgi*):c.331A>C are a case in point. We found significant associations with larval development. The AA homozygotes have heavier larvae following diapause, but smaller pupae than the other genotypes. These results were consistent across the four populations, even though there are great differences in the frequency of the C allele in the populations: the C allele is much more common in the two northern populations from continuous environment (around 50%) than in the southern populations from fragmented environment (around 20%).

Several previous studies conducted in the Åland Islands have demonstrated that the (*Pgi*):c.331A>C AC heterozygotes have superior flight metabolic rate and dispersal rate in the field compared with the AA homozygotes (*Niitepõld et al., 2011*, *2009*). Both empirical and modeling studies have suggested that natural selection in highly fragmented landscapes favors heterozygosity because of associated superior dispersal and colonization abilities (*Fountain et al., 2016*; *Hanski & Mononen, 2011*), thus allowing the individuals to differently cope with spatial heterogeneity, and the species to best occupy the available habitat. Similarly, other phenotypic traits (e.g., larval and fecundity) may be affected by the prevailing environmental conditions (*Saastamoinen, 2007b*; *Saastamoinen et al., 2013*). Working on larval development, *Saastamoinen et al. (2013)* found that individuals with the C allele develop more often via an extra larval instar before pupation. As the extra instar is common in larvae that are small after diapause, this finding is consistent with our results showing that the AA larvae are heavy after diapause, as well as with results from *Kallioniemi & Hanski (2011)* showing that AA homozygotes produce heavier pupae than the other genotypes. Furthermore, the homozygote AA individuals have higher survival in low (stressful) temperature in the laboratory (*Kallioniemi & Hanski, 2011*), while the small CC individuals generally showed reduced survival (*Orsini et al., 2009*). These results are consistent with studies on other butterfly species. In the copper butterflies, *Lycaena tityrus*, *Karl, Schmitt & Fischer (2008)* found that *Pgi* genotypes are associated to several traits of larval and pupal

development and cold stress resistance, while in *C. eurytheme Pgi* heterozygote individuals have higher survival than the homozygotes in natural populations (*Watt, 1977*).

Although, the current study did not highlight any genetic and environmental association with female adult traits (including survival, fecundity, and flight activity traits), previous studies have shown that *Pgi*-AC heterozygous females (SNP (*Pgi*): c.331A>C) lay larger clutches than AA homozygous females especially at low ambient temperatures (*Saastamoinen, 2007a*; *Saastamoinen & Hanski, 2008*). In *Colias*, *Pgi* heterozygous individuals in allozyme studies had similarly high male mating success and high female fecundity (*Carter, 1988*; *Watt, 1992*; *Watt et al., 2003*). The CC homozygous individuals are uncommon in the Åland Islands and hence previously little studied. We suggest that this lack of association with female traits may be due to the large number of traits included in the PCA, rather than the true lack of association with genetic polymorphism in the two environment types.

In summary, several studies on the Glanville fritillary examining associations between the SNP (*Pgi*):c.331A>C and life-history traits have found consistent results for several traits, including larval development. The individuals with the AA genotype do not appear to be as good dispersers as the heterozygotes (*Niitepõld et al., 2009*), but might re-allocate their resources towards other fitness traits, such as their own body size, making them competitive individuals within their habitat patches (e.g., in many butterflies, females select for larger males) and the environmental conditions (e.g., fluctuating temperatures). There may also be trade-offs between dispersal behavior and immune responses, as suggested by the significant associations observed between another gene, the *Heat shock 70 kDa protein* gene, and adult male activity (PC$_{M3}$). Active flight causes oxidative stress, which may become an important factor for dispersive butterflies. *Somervuo et al. (2014)* showed that butterflies from fragmented landscapes are more tolerant of hypoxia, which is suggested to be another likely genomic adaptation to dispersal.

## CONCLUSION

The current literature shows multiple evidence pointing to the possibility that population-level variations in the environmental conditions, in the Baltic Sea region, may influence life-history evolution and thereby allele frequencies and gene expression in the Glanville fritillary butterfly. In the context of habitat fragmentation, traits such as dispersal and re-colonization, which are key processes facilitating the persistence of metapopulations in fragmented landscapes, are among the traits that have previously been suggested to be affected (*Hanski, 2011*). Equally, traits influencing performance in large, stable populations (in continuous landscapes) may be important. Studies on allelic variation in *Pgi* and *sdhd* genes have similarly demonstrated significant differences between new versus old local populations in the highly fragmented landscape in the Åland metapopulation (*Hanski & Saccheri, 2006*; *Hanski, 2011*; *Wheat et al., 2011*; *Zheng, Ovaskainen & Hanski, 2009*), supporting the notion that population turnover, extinctions and re-colonizations, impose selection in fragmented landscapes. Similarly, other gene expression studies

demonstrated significant allelic differences in various genes between individuals experiencing different ambient temperature conditions (*Kvist et al., 2013*; *Saastamoinen & Hanski, 2008*; *Saastamoinen et al., 2013*). The differences between newly established versus old local populations in fragmented landscapes, and between individuals evolving in cold versus warmer environments reflect some of the selection processes that might operate at the population level. Although we could predict that the difference between environments should reveal the outcome of such processes during longer periods of time, further functional analyses are needed to fully tease apart the relative importance of the confounding and stochastic evolutionary processes related to the degree of habitat fragmentation and the geographical positions of the five populations used in our study.

## ACKNOWLEDGEMENTS

We thank V. Ahola, G. Blanchet, SW. Geange, P. Somervuo, A. Tack, T. Schulz, and E. Numminen for help with the analyses, M. Saastamoinen for precious discussions on the topic of the study, and H. Körnich for providing us with the proper SMHI weather data. Suvi Ikonen and all the assistants at the Lammi Biological Station are thanked for help with data collection. The authors want to dedicate this manuscript to the memory of Prof. I. Hanski for his supportive leadership and numerous major contributions to Ecology.

### Funding

The project was funded by the Academy of Finland grant #250444 and #256453 to IH, and #266021 to AD, as well as by the European Research Council (AdG grant #232826 to IH). The funders had no role in study design, data collection and analysis, decision to publish, or preparation of the manuscript.

### Grant Disclosures

The following grant information was disclosed by the authors:
Academy of Finland: #250444, #256453 and #266021.
European Research Council (AdG): #232826.

### Competing Interests

The authors declare that they have no competing interests.

### Author Contributions

- Anne Duplouy performed the experiments, analyzed the data, wrote the paper, prepared figures and/or tables, and reviewed drafts of the paper.
- Swee C. Wong analyzed the data, prepared figures and/or tables, and reviewed drafts of the paper.
- Jukka Corander analyzed the data and reviewed drafts of the paper.
- Rainer Lehtonen conceived and designed the experiments and reviewed drafts of the paper.
- Ilkka Hanski conceived and designed the experiments, wrote the paper, and reviewed drafts of the paper.

## Data Availability

Dryad: DOI 10.5061/dryad.81r20; https://datadryad.org

The accession number will only be available once the paper has been accepted for publication in PeerJ. In the mean time I have provided the reviewers with the raw data (three excel files only for review) in the supplementary files of the submission.

## Supplemental Information

Supplemental information for this article can be found online at http://dx.doi.org/10.7717/peerj.3371#supplemental-information.

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
