# Peer review of "Genetic effects on life-history traits in the Glanville fritillary butterfly"

_PeerJ, doi:10.7717/peerj.3371_

## Round 0.1 · original submission · Major Revisions

You were fortunate to get two excellent, careful reviewers, both of whom asked for revisions in a constructive way. I concur with the reviewers, and call for major revision of the MS, according to their comments. Both reviewers want you to re-work the introduction and to explain your methods more completely. Both pointed out that environmental fragmentation and latitude are confounded in your study; you should therefore discuss this problem prominently.

Like Reviewer 1, I want you to spell out how you avoided Type I errors (too many “significant” associations). I’m not a statistician, but I wonder if the Benjamini-Hochberg approach to avoiding false positives would be applicable. Reviewer 2 has the opposite concern: false negatives due to small sample sizes. Here, the thing to do is discuss this possibility in the Discussion section.

The reviewers ask you to make a lot of changes, but I view this as a positive thing, as your aim is to make your paper as good as possible. Please follow the reviewers’ advice, write a point-by-point list of your changes, and re-submit the Ms to PeerJ.

Reviewer 1 ·

Basic reporting

The manuscript was in professional article structure with clear and unambiguous English throughout. The raw data was provided and will be available on data dryad.

The introduction could be developed more to layout specific question being addressed and hypotheses that are being tested. I would begin with a more general introduction that expands on the ideas presented in the “background” section of the abstract to address the specific question being asked before getting into what is already known about specific genes associated with metabolism in butterflies. I would start the final paragraph of the introduction with the hypotheses and then include the description of what your data show.

Experimental design

My biggest concern is that there were so many traits (25 Life history traits and 4 PC of the original variables) and SNPs (49 SNPS from 26 genes) being tested in this project. Given the number of SNPs tested and the multiple forms of inheritance tested (additive, dominant, recessive and over-recessive/dominant [lines 242], it is unclear if there are more significant associations than would occur by chance. The authors state that they corrected for multiple comparisons [e.g. lines 245-246 each raw p-value was corrected for a maximum of 36 tests (36 SNPs tested) conducted on each trait individually], but it is not clear how this was done. It seems like traits were analyzed and treated differently without a clear reason why, with some examples listed below:

A. Using the PC for the larval traits made sense as many of these traits may be correlated. It was unclear why it was not done consistently across the data (e.g., Fig 1 A is looking at PC3 and Fig 1B is 5th instar weights). I also think the adult data might benefit from such analysis as many of these traits 21 phenotypic adult traits are probably also correlated.

B. [228-230] For traits related to female fecundity, such as clutch size, lifetime corrected number of eggs or age at 1st oviposition, we included female ID as a random factor nested within population.
It seems that ID should be a factor in all of the models, so why just fecundity?

C. [234-235] Clutch size and clutch rank are negatively correlated and hence we included clutch size only in some models
How was it determined which models you included clutch size and which did not?

D. [247] Correlation between SNPs was not taken into account
Since the SNPs are quantified for the same individuals, calculating linkage disequilibrium among the SNPs would prevent treating linked SNPs as independent. Based on the description on LD variation across the genome [line 251-253] it is possible that some of the SNPs in the same gene are in the same haplotype block and should not be treated independently.

E. Supplemental Table 1 is described to include all of the results instead of just the significant results presented in table 1, but I find it very confusing to read. I think each row represents all of the SNPs that were tested for that trait. I think it would be clearer if each test is a separate row.

Overall, I think the methods would benefit from a more clear explanation as to what comparisons were tested and what corrections were made for so many tests.


A second consideration is the focus on characterizing these environments into fragmented and continuous landscapes. These habitat characterizations also correspond with differences in latitude and the authors note that habitat fragmentation and latitude are confounded [line 90]. It is not clear why the focus remains on the fragmented versus continuous characterizations as there are other environmental parameters that are clearly different between these populations, notably temperature; Figure 5b shows that the high-latitude fragmented populations have lower minimum and average temperatures across the year. It is also unclear why the data is displayed by population in Figure 1 but by habitat type in all other figures.

Validity of the findings

The discussion did a nice job putting the results in the context of the broader base of literature.
The authors speculated on the broader implications without overstating the significance of their results and pointed towards specific questions that required further investigation. However, given the number of genes and traits being discussed, the discussion became difficult to follow at times jumping from one gene to the next. The list of genes and function [lines 483-504] helped create a structure and might be beneficial earlier in the discussion.

I again found the conclusions based on habitat type (continuous versus fragmented) to make assumptions that this trait is driving the differences among populations. I appreciated the discussion of alternative population characteristics such as historical relatedness [lines 473-478], however, this could be addressed using neutral markers. The authors do not discuss environmental parameters beyond habitat fragmentation that vary among these sites.

Additional comments

Summary:
This paper analyzed the samples that had been collected in a phenotypic study (Deplouy et al 2013) genetic associations with candidate SNPs metabolic genes. They examine differences across 5 populations and include data from a pilot study for semi-temporal replication. The authors found that several of the SNPs tested correlated with some of the life history traits that were assessed. It is exciting to look at function of natural polymorphisms in shaping local adaptation in natural populations of a non-model organism. My biggest concern with the paper was the clarity of the statistics involved to correct for so many comparisons being tested.

Reviewer 2 ·

Basic reporting

Abstract:
22-25: It would be beneficial to add the main question to the Background part, e.g. by mentioning that the genetic basis of life-history traits and how habitat fragmentation affects these traits is poorly understood.
26-28: Please give more details to the association study, e.g. by mentioning that variation in several life-history traits was associated with several factors (including genetic SNP markers). Also please include the number of markers used.
38: I recommend rephrasing the sentence to: “within the population from the Aland islands”

Introduction:
General: The introduction covers most of the topics relevant to this study, but it could be improved by adding more information on habitat fragmentation. For instance, how does habitat fragmentation affect life-history traits and what are the actual selective pressures? Please make it clearer why the Glanville fritillary is a good model to study habitat fragmentation in terms of genetic and phenotypic evolution.

76: The phrase “selected loci” might be misleading. I understand that the authors selected these loci, but it could also be understood in terms of that these loci are subject to natural selection. I recommend rephrasing.
78: Please explain how the other studies were “more limited” (in terms of what compared to this study?).

Methods:
General: The Data analyses part of the Methods is difficult to read and not substantial enough for the reader to understand it entirely. This is a major problem that needs improvement. I recommend, for instance, to add equations of the linear models used, instead of just explaining them in writing. It is unclear which factors were fit for which traits, and the levels of the factors need to be clearly mentioned in the text. For instance, it is not clear which interactions were fit. There is not enough information to determine this in the text. I also recommend the authors to do a model selection, as the models with 3 factors + interactions might have been overfit.

101: Please mention: based on which analysis and cut-offs were traits closely correlated to the selected traits excluded? The selected traits are to some degree strongly correlated as well, as suggested by the PCA.
105: Should that be “five” instead of “four” populations? The figure and text (107-113) indicate that five populations were sampled.
122-131: It is not entirely clear that there were 2 studies in different years and what they were used for exactly. Please explain this briefly before going into details.
127: “were most likely unrelated to each other” is quite vague. I recommend rephrasing to “did not have a high degree of relatedness” and mention what analysis revealed this.
154: Please mention how “sunny day” was defined. Within how many days did the 10 sunny days occur?
191: Please mention the type of markers (e.g. 31 SNP markers)
203: Please add “(HW equilibrium)” next to Hardy-Weinberg equilibrium. You are referring to the HW equilibrium in the text below, without defining the abbreviation at this point.
210: Please mention whether or not only bi-allelic SNPs were included in this study.
217: Please mention the version of R used and make sure that the reference is correct.
218: How was independence between response variables determined? This is unclear.
219: Please give more details on how the normality tests were performed. Which tests and P-value cut-offs were used to determine normality? Which transformations were used for which traits – and did this improve normality?
242: Please explain how you accounted exactly for additive, dominant, recessive, etc. effects
243: I believe that “false recovery rate” should read “false discovery rate”.
261: Please change “ps” to “p-values” throughout the manuscript.
273: The R package needs to be correctly referenced.
281: Temperature data were from years different to the sample collection years. Please indicate whether this is still representative. Also, please explain how these temperature data were used in the present study. Have temperature data been used in any statistical models? The purpose of showing these at all (in Figure 6B) needs to be defined better.

Results:
General: The sample sizes used to determine the effect of SNPs on genotypes are quite small. I believe that this leads to a lot of false negatives, which the authors have not identified – and this should be made clear also in the discussion. Furthermore, even though the authors mention that “sex” and “landscape” were included in the models, they do not mention the effect of these factors on the different traits adequately in the results (or discussion) – please improve this. Moreover, the allele frequency differences between fragmented and continuous landscapes are confounded by the distance these populations are apart. Allele frequency differences can also be explained through isolation by distance, and I would recommend checking if the frequency differences are more likely due to distance or different habitats. It would further be interesting if the authors could show a more comprehensive analysis on the genetic differences between populations in addition to the heatmap, for instance, through a structure or FST analysis.

292: Please change “detailed results” to “details on statistical results”.
298: Please make sure that the “P” for p-values is either consistently written in capital or small throughout the manuscript.
309: An interaction between SNP and landscape indicates that the SNP depends on the genetic background. This can be explained by linkage and/or epistasis. Please make sure that this is clearly stated (in results or discussion).
323: Brackets are missing in the text.
361: There is a typo in SNPS, which should read “SNPs”

Discussion & Conclusion:
General: I believe that not much conclusions can be drawn in terms of how habitat fragmentation affects these populations using this data set. The main reason for this is that, here, habitat fragmentation is confounded with physical distance and several varying environmental factors. The authors did not manage to resolve this soundly enough. The genetic associations with many phenotypes are interesting, though it is not entirely clear from the methods how they were obtained exactly. They are not novel to a large degree, but it is good that there is a lot of independent evidence that these genes are involved in life-history traits. Please label figures with multiple panels with A, B, C, etc.

484: What does “2-14” refer to?

Figures and Tables:
Please number figures and tables according to their occurrence in the text. For instance, Figure 6 is mentioned very early in the text, and this should be re-labelled to Figure 1. This is a very informative Figure to the reader as it shows the studied populations! It would be good to have water and land in different colors (e.g. water in grey and land in white). Please also remove the sample sizes from the Figures and put them in a separate table with all populations and traits. Please also move the temperature panel of Figure 6 to the supplement, as it is not really discussed in the text. Moreover, please improve the legends and give more details – particularly on the abbreviations. This mainly applies to the supplementary material. In Table S2 it is further not clear what the numbers refer to (Pearson’s correlation?) – please give more details on this and why some numbers are in bold.

Experimental design

This is an interesting study analyzing how habitat fragmentation affects life-history on the phenotypic and genetic level. Phenotypic assays are well designed, but the statistical analysis can be improved (see basic reporting). In addition, the methods – in particular the data analysis part – are poorly described and require substantial improvement. While the genetic associations are well supported through independent evidence from other studies, there are likely many false negatives due to small sample sizes used here. Moreover, as a consequence of confounding factors (e.g. environment) it is difficult to infer whether genetic differences are caused by habitat fragmentation.

Validity of the findings

It is difficult to determine if the statistics used is appropriate, as the methods are not having sufficient information. Also, even though multiple factors were analyzed in the models, not all of them are reported. This is however crucial to the reader to get a complete picture on the results. Please see “basic reporting” part of the review for more.

Additional comments

This is an interesting study, but needs substantial improvement in how the methods are explained and the results reported. The authors did a lot of phenotypic characterization and used several statistical models here, and this is therefore a lot of information for the reader. The manuscript generally requires improvements in clarity, explanations of the methods and reported results.

---

## Round 0.2 · Minor Revisions

I ask for a few truly minor changes:

lines 399 - 400: instead of "Cytochrome P450" please write "CYP337", and instead of "(HSP)" please write "(Hsp70)"

lines 468-494: Each gene listed should be preceded by "the". Begin
line 468 "2. The Cytochrome P450 gene CYP337,". Begin line 474 " 4. The Heat shock protein gene Hsp70,"

I think you answered the reviewers criticisms. Too bad none of the genotype-PC associations came out "significant" by the Benyamini analysis.

If you could perform a power analysis that would suggest sample sizes required to detect associations, that would be a very useful addition to the paper. If not, let it go.

---

## Round 0.3 · accepted · Accept

Anne, thanks for your rapid response.
All the best,
John